# Mental imagery and breathing exercises integrated into a standardized warm-up routine enhance sympathetic activation and optimize muscular performance in firefighters

Jean Philippe Biéchy[1,2], Camille Charissou[1,2], Candice Groléas[3], Thomas Skrysinski[3], Sylvain Gobert[4], David Amarantini🔴[1�], Lilian Fautrelle🔴[1,2�]

**1** ToNIC, Toulouse NeuroImaging Center, UMR 1214, Université de Toulouse, INSERM, Toulouse, France, **2** Institut National Universitaire Champollion, EIAP, Département STAPS, Campus de Rodez, Rodez, France, **3** SDIS 42, Service Départemental d'Incendie et de Secours de la Loire, St-Etienne, France, **4** SDIS77, Service Départemental d'Incendie et de Secours de Seine et Marne, Melun, France

☯ These authors have contributed equally to this work and share last authorship.
* david.amarantini@inserm.fr

## Abstract

The capacity of firefighters to consistently mobilize their full physical potential during duty shifts is of paramount importance. One promising approach to achieving this goal involves developing operational protocols that effectively activate the sympathetic nervous system and vagal regulation prior to each professional action. This study investigated the effects of combining mental imagery and conscious breathing exercises with a conventional and standard warm-up on cardiac modulations, muscular strength and endurance performance in 34 firefighters randomly assigned to one of two groups: a control group (n = 17), which performed only the conventional and standard warm-up, and an experimental group (n = 17), which performed the standard warm-up in combination with mental imagery and breathing exercises. The results showed that incorporating such psychophysiological techniques into a conventional warm-up routine during repeated physical efforts optimizes sympathetic spectral power, heart rate responses, and physical performances, including maximal voluntary isometric handgrip contraction, maximum push-ups, and the maximal abdominal plank duration. Altogether, these findings indicate that brief mental imagery and controlled breathing exercises incorporated into standard warm-up routines effectively preserve muscular performance under repeated exertion while enhancing sympathetic activation, offering a practical and immediately applicable strategy to optimize firefighters' operational readiness.

**Data availability statement:** All relevant data are within the manuscript and its Supporting information files.

**Funding:** The author(s) received no specific funding for this work.

**Competing interests:** The authors have declared that no competing interests exist.

## 1. Introduction

Firefighters are required to operate in hostile environments that involve multiple highly demanding physical tasks combined with highly engaging emotional contexts, both of which elicit substantial cardiovascular strain – characterized by pronounced elevations in heart rate – while rescuing victims and combating fires, often at the risk of their own lives [1]. A high level of physical fitness, muscular endurance and muscular strength has been associated with improved performance in operational firefighting tasks – such as victim rescue, hose handling, ladder climbing, and equipment carrying – as well as a reduced risk of injury [2–4]. Therefore, the ability of firefighters to consistently maximize their physical capabilities throughout their duty shifts is of critical importance.

Moreover, several factors during the 48- to 72-hours on-call duty period contribute to significant fatigue: the repetition of physically demanding interventions and/or exposure to high levels of psychological and physiological stress [5], circadian variations [6], repeated handling of heavy objects [5], psychological workload [7] and emotionally intense interactions [8] have all been shown to decrease muscle strength and fitness performance. Mitigating these effects of fatigue is thus a key challenge, as they jeopardize mission success, health and safety by increasing the risk of accidents [9] and reducing work endurance time by up to 75% [10].

From a neurophysiological perspective, muscular strength refers to the capacity to generate force during a single, brief, maximal voluntary contraction of a muscle at rest [11], whereas dynamic and isometric muscular endurance can be defined as the ability to sustain or repeat muscular effort across multiple dynamic movements or during prolonged postural maintenance [12]. Importantly, these physical capacities are governed not only by peripheral musculoskeletal factors but are also strongly modulated by neural mechanisms, including the central nervous system (CNS) and the autonomic nervous system (ANS), which regulates heart rate (HR) and heart rate variability (HRV). This neurophysiological regulation plays a crucial role in determining how fatigue develops and how performance is maintained under operational constraints [13]. In this context, activating the sympathetic nervous system prior to strenuous effort enhances cardiovascular readiness and facilitates greater muscle recruitment, thereby improving strength and endurance performance [14,15]. Such pre-activation primes the body to meet acute energetic demands by increasing heart rate, blood flow, and neuromuscular excitability, thereby potentially optimizing warm-up efficacy.

Within this framework, and without modifying firefighters' work schedules (in France, in the present study), two potential countermeasures could help prevent and counteract the detrimental effects of fatigue during on-call duty. First, implementing operational protocols to enhance recovery during intermittent breaks between interventions has been proposed [16]. Second, developing protocols that stimulate firefighters to optimize their performance by activating the sympathetic system and modulating the vagal brake -irrespective of their fatigue level- may be beneficial. In pursuit of this goal, alternative strategies beyond traditional warm-up routines, which are the standard preparation method among French firefighters [17], have been explored to enhance physical readiness and optimize muscular performance.

Firstly, certain mental training techniques, such as motor imagery [18] and emotional visualization [19], have been identified as potential methods to enhance corticospinal excitability, thereby promoting muscle strength and endurance. Although the primary and well-established effects of motor imagery are neural and centrally mediated, several studies also suggested that it can influence autonomic responses, as evidenced by changes in heart rate and heart rate variability [20,21]. Motor imagery involves the mental simulation of a muscle action or contraction without actual movement execution [22]. During mental imagery, individuals engage neural circuits similar to those activated during real movement execution [23]. Consequently, this process increases corticospinal excitability compared to rest [24,25], leading to greater muscle activations [26], enhanced muscle strength [27], and delayed neuromuscular fatigue onset [28].

Secondly, another potential mechanism to stimulate the ANS is through controlled breathing exercises [29], which are defined as voluntary modifications of respiratory patterns, distinct from autonomic breathing regulation in daily life. Depending on the duration and nature of inhalation and exhalation phases, breathing exercises can either enhance sympathetic or parasympathetic activity, thereby either invigorating or calming the body [30]. Specifically, stimulating sympathetic activation before intense physical exertion may facilitate cardiovascular preparation, improving muscular strength and endurance performance [14,15].

In this regard, the present study aimed to examine the impact of combining mental imagery and conscious breathing exercises with a standardized warm-up routine on cardiac adaptations and subsequent muscle strength and muscle endurance performance in French firefighters. We hypothesized that firefighters performing a standardized warm-up routine supplemented with mental imagery and breathing exercises would exhibit less performance decline – or even performance enhancement – during repeated maximal efforts, including maximal voluntary isometric handgrip contraction, maximum number of push-ups, and maximal abdominal plank duration. These practices were expected to help counteract both central and peripheral fatigue while enhancing corticospinal excitability, cardiovascular readiness and sympathetic activation, compared with controls who performed only the standardized warm-up routine without psychophysiological techniques. To test this hypothesis, the study utilized HRV measurements, which analyze the R-R interval fluctuations and represent one of the most widely used non-invasive metrics to assess ANS function. HRV specifically provides insights into the sympathetic (the body's "energizing" system) and parasympathetic (the body's "braking" system) branches of the ANS [13,30].

## 2. Materials and methods

### 2.1 Participants

Based on G*Power calculations (f = 0.25–0.30), α = 0.05, and power = 0.80, a total of [28:32] participants was required, consistent with prior psychophysiological studies [16,31]. We therefore recruited 34 firefighters (age: 34.9 ± 6.9 years; height: 178 ± 4.2 cm; weight: 77.4 ± 5.8 kg; body mass index: 24.6 ± 1.1 kg/m2) volunteered for the experiment. Participants were not selected based on specific fitness criteria. Eligibility requirements are detailed in Appendix 1. Firefighters who did not meet at least one of the 11 eligibility criteria were excluded from the study. No explicit information regarding the study's objective was provided to participants prior to the experiment. Recruitment was conducted from April 1 to July 30, 2021. All participants provided written informed consent before taking part in the study. All procedures were conducted in accordance with the 2008 Declaration of Helsinki and received approval from the Ethics Committee of the Université Fédérale de Toulouse Midi-Pyrénées (Approval No.: 2020-04-21-213).

Participants were then randomly assigned to one of two groups (n = 17 per group): a control group (CTRL), which performed only the conventional and standardized warm-up routine, and an experimental group (EXP), which performed exactly the same the conventional and standardized warm-up routine within which was integrated a combination of mental imagery and breathing exercises (see Table 1 for details).

### 2.2 Experimental protocol

The complete experimental diagram and procedure are presented in Fig 1.

**Table 1. The warm-up contents step by step from the first until the thirteen minutes (left column) for the CTRL (middle column) and the EXP group (right column). The EXP warm-up contents are composed by the same sequence of real muscle contractions as the CTRL group, to which are integrated, during the performance of the muscular exercises, the mental imagery practices and the breathing exercises that are specifically indicated in the right column.**

| Timeline of the warm-up (min) | Instructions and tasks for the CTRL group | Additional instructions and additional tasks for the EXP group |
|---|---|---|
| 0:00-1:00 | 5 neck rotations in the antero-posterior plan (the "yes" movement), 5 left-right/right-left rotations (the "no" movement), 10 full head circles (5 clockwise and 5 counter clockwise). | Neck rotations + conscious breathing: during the neck rotation exercises, participants were asked to voluntarily controlled their breathing. They were instructed and guided to fully inhale 3 seconds through their nose and exhale sharply through their mouth in one second. |
| 1:00-3:00 | 5 forward followed by 5 backward arm rotations for each arm alone then both together, and ended with 20 shrug movements. This one-minute cycle was performed twice. | Arms rotations + energizing breaths + mental imagery: participants were asked to continue the same energizing breaths than before. Moreover, they were instructed to imagine being doing push-ups with strength and vigor as if they had a weight on their backs, then the heavy weight than they could raise, and to feel "the sensations of muscle tension and powerful strength generated to extend arms". |
| 3:00-5:00 | Arm swings: participants hold their arms out to the side at shoulder height, circle forward around their arms slowly, starting with small circle and working up to largest circles in 20 repetitions, and then performed 10 more movements of large circles with their arms swinging forward. The same sequence was then performed in a backward way. | Arm swings + mental imagery: during the arm swings, participants had to imagine to feel sharp tension in their arms muscles, almost painful, and visualize themselves to be breaking their push-up record with a sense of pride. |
| 5:00-7:00 | Isometric half squat: from a standing position, participants bend their knees until 45°, maintained this position 20 seconds et then go back to the initial position (4 repetitions in two minutes). | Isometric half squat + empty lung apnea: participants had to perform their isometric half squats during an empty lung apnea and inhale fully when go back to the initial position. |
| 7:00-9:00 | Full body concentric contractions: 10 calf raises, 5 half squats, 5 full squats, 5 push-ups, 5 burpees (one full sequence every minute). | Full body concentric contractions + mental imagery – empty lung apnea: during all the duration of these cross-fit movement executions, participants were asked and guided to imagine to "feel sharp tension in all the muscles of their arms, trunk, and legs, almost painful as if they were at the end of the abdominal plank test, and visualize themselves to be breaking their abdominal plank maximal record with a wide sense of pride" |
| 9:00-11:00 | Hopping in place: arms contracted downwards, hopping in place during 30 seconds followed by a 30 seconds recovery period, twice. | Hopping in place + full lung apnea: every period of hopping in place was realized with a full lung apnea. |
| 11:00-13:00 | Torso lateral flexions and torso twists: from a standing posture, from slow to natural speed and from small to maximal amplitude, 20 torso lateral flexion and 20 torso twists (10 rights, 10 lefts). | Torso lateral flexions and torso twists + energizing breaths + mental imagery: during the torso lateral flexion and twist exercises, participants were instructed and guided to fully inhale 3 seconds through their nose and exhale sharply through their mouth in one second. In the same time, they were asked to visualize themselves in the present environment, feel the muscular warmth of their muscles… visualize the fully oxygenated blood flowing abundantly into their powerful muscles of their legs, trunk, arms, and now feel fully ready to perform their second fitness tests series. |

The experiment was conducted in a controlled environment, with participants wearing sports clothing inside the fire station gymnasium. Key environmental and physiological factors, including test timing, hydration, humidity, ambient temperature, as well as meal and restroom breaks, were standardized and maintained consistently for all participants throughout the experiment.

The protocol began with a PRE-test session in which the maximal voluntary isometric contraction (MVIC) of the handgrip, push-ups, and abdominal plank were assessed for each participant. Following this assessment, participants underwent a 30-minute rest period in a passive standing posture on both feet, during which all parameters potentially

## Experimental design

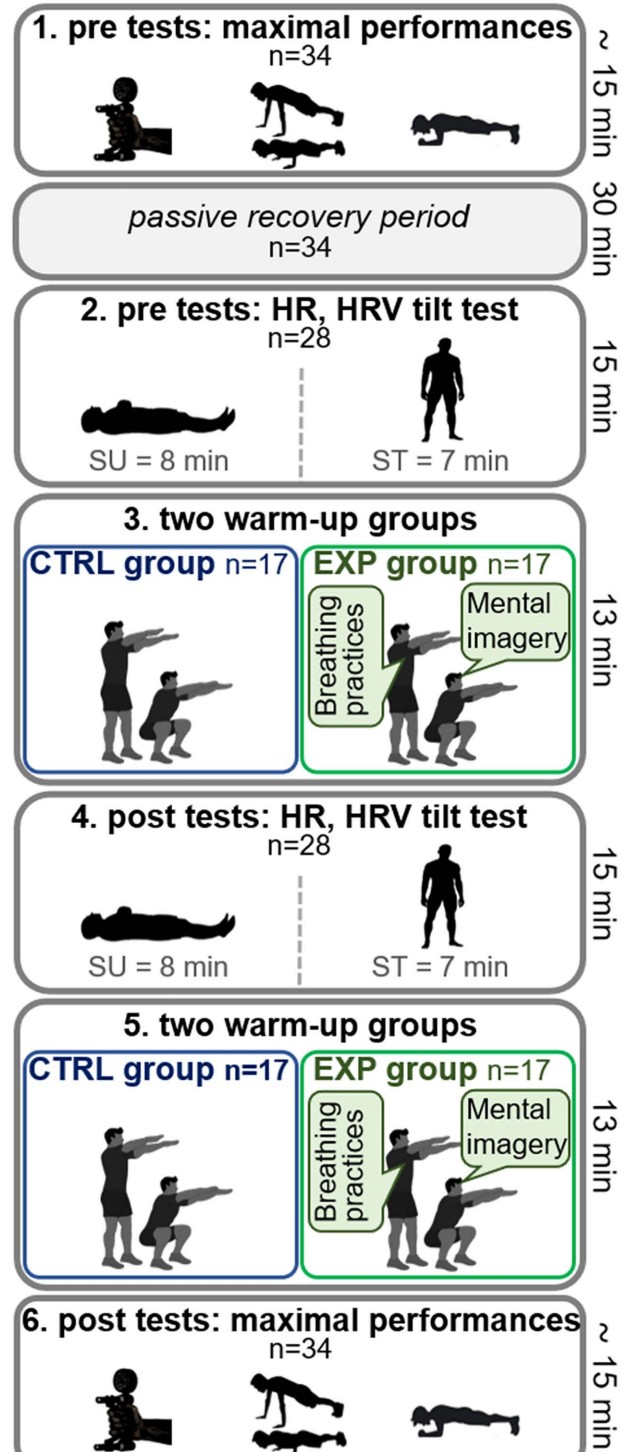

**Fig 1. Experimental design.**

influencing autonomic function (hydration, restroom use, posture) were controlled and standardized. During this interval 28 out of the 34 firefighters were randomly selected to be fitted with a heart rate monitor (14 per group; S810i, Polar, Aulnay-sous-Bois, France). At the end of the rest period, an HRV tilt PRE-test was performed, following the methodology previously described by our research team [16] and other studies [13].

Immediately after the assigned warm-up, the same HRV tilt test administered in the PRE-test session was repeated POST-test for each participant. Since the tilt test is a non-exertional procedure lasting a total of 15 minutes, participants completed a second round of their respective warm-up protocol before proceeding to the POST-test session for maximal muscular performance assessment. Finally, the same muscle performance tests conducted during the PRE-test session (i.e., handgrip MVIC, push-ups, and abdominal plank) under identical environmental conditions were repeated POST-test for each participant.

## 2.3 The CTRL and EXP warm-up

The CTRL and EXP groups completed a standardized 13-minute warm-up routine commonly implemented by firefighters in the Loire department (France) (see Table A1, middle column). The warm-up was conducted in a temperature-controlled environment set at 24°C (±0.5°C) and was supervised by two professional sports instructors. In addition, the EXP group performed a combination of mental imagery and conscious breathing exercises integrated into the warm-up protocol (see Table 1, right column).

## 2.4 Muscular endurance and strength tests

Three assessments were conducted to quantify muscular endurance and strength performance:

- **The MVIC hand grip test:** the first test measured maximal voluntary isometric contraction (MVIC) of the hand grip, following the grip strength assessment protocol established by the American Society of Hand Therapists [32,33]. As recommended by Bechtol [34], a calibrated dynamometer with adjustable grip spacing (model: 200lbs/90 kg 5030J1, Jamar, Sammons Preston Rolyan, Nottinghamshire, UK), was used to ensure measurement accuracy, in accordance with calibration procedures outlined by Fess [35].

- **The maximum push-ups:** the second assessment measured the maximum number of push-up repetitions, as used in the selection and aptitude test for French firefighters. Participants started in a standardized position: feet placed 10 cm apart, weight supported on the toes, arms fully extended, hands shoulder-width apart, and the body maintaining a straight alignment (head, trunk, pelvis, knees, and feet forming a single line). During each repetition, participants lowered themselves until their chest made light contact with the ground without bouncing, while maintaining body alignment. Participants were not permitted to remain motionless for more than two seconds in either the starting or lowered position and were instructed to perform as many push-ups as possible. Testing was terminated when participants could no longer meet the performance criteria, with the last correctly executed repetition recorded. A six-minute passive recovery period was observed before proceeding to the third test.

- **The maximum duration of abdominal plank:** the third assessment evaluated core endurance using the abdominal plank test, another standard component of the French firefighter selection and physical aptitude test. Participants assumed the plank position with their forearms and feet on the ground, ensuring full-body alignment (head, trunk, pelvis, knees, and feet forming a straight line). The stopwatch (Vantage V2, Polar, Aulnay-sous-Bois, France) was activated as soon as participants assumed the correct position and was stopped when they could no longer maintain proper form.

## 2.5 Borg Rating of Perceived Exertion (RPE)

Immediately after completing the final muscular performance test in both the PRE-test and POST-test phases, subjective effort perception was assessed using the 10-point Borg Rating of Perceived Exertion (RPE) scale [36].

## 2.6 HRV recordings and analyses

HRV recordings and analyses strictly adhered to the methods and procedures described by Biéchy et al. [16]. In brief, heart rate (HR) and heart rate variability (HRV) were recorded during standardized tilt tests, following established recommendations [13]: participants remained in a supine position (SU) for 8 minutes, followed by 7 minutes in a standing position (ST).

To avoid the potential bias introduced by a very low respiratory rate (< 6 cycles per minute) in the supine position [37], which could affect HRV interpretation, investigators visually monitored participants' respiratory rates to ensure they remained above 6 cycles per minute. No instances of low respiratory rates were observed.

HRV analyses were conducted using R-R interval data, with recordings extracted from the 3rd to the 8th minute in the SU position, and from the 9th to the 14th minute in the ST position, as recommended by Weippert et al. [38]. Each recording segment lasted 256 seconds, yielding 512 data points after resampling at 2 Hz (Kubios HRV Premium software, version 3.5, [39]).

The power spectral density was calculated using fast Fourier transform [40], and HRV parameters were analyzed in both the time and frequency domains, following the recommendations of [41]. The following spectral components were assessed: (i) LF (Low Fréquency, a marker of both sympathetic and parasympathetic modulation), HF (High Fréquency, a marker of parasympathetic modulation), total spectral power (TP = LF + HF), and the LF/HF ratio in SU and ST positions. All values were expressed in absolute spectral power units (sec²/Hz).

## 2.7 Statistical analysis

All data are presented as mean ± standard deviation. Each dependent variable data set followed a normal distribution (Shapiro-Wilk test, average $p > 0.16$) and met the assumption of sphericity (Mauchly's test, average $p > 0.27$). The effects of incorporating mental imagery and breathing exercises into a standardized warm-up routine were analyzed by comparing POST-training measures of hand-grip MVIC, maximum push-ups, maximum abdominal plank duration, HR, and HRV parameters using mixed-design 2 × 2 ANOVAs. Session (PRE vs. POST) was treated as a within-subject factor, while Group (CTRL vs. EXP) was treated as a between-subject factor. When significant Session × Group interactions were observed, post hoc analyses were conducted using Scheffe's test where appropriate. Effect sizes ($\eta^2$) were reported, with thresholds for small, moderate, and large effects set at 0.2, 0.5, and 0.8, respectively [42].

## 3. Results

### 3.1 Muscular performances

**3.1.1 MVIC of the handgrip (Fig 2A).** The ANOVA revealed no main effect of Group ($F_{(1,34)} < 0.005$, $p = 0.99$, $\eta^2 < 0.001$), a significant Session effect ($F_{(1, 34)} = 31.9$, $p < 0.05$, $\eta^2 = 0.21$), and a significant Group×Session interaction effect ($F_{(1, 34)} = 32.8$, $p < 0.001$, $\eta^2 = 0.61$). No significant difference in the MVIC of the handgrip was observed between the CTRL and the EXP groups in the PRE-test phase (57.4 ± 9.9 and 53.4 ± 9.3 kg in average for the CTRL and the EXP group respectively, $p = 0.24$, Scheffe post-hoc). By contrast, POST-test handgrip MVIC was significantly higher in the EXP group compared to the CTRL group (55.7 ± 8.9 vs. 52.3 ± 9.2 kg, $p = 0.04$, Scheffé post hoc). The Group×Session interaction was due to the fact that the MVIC of the handgrip in POST compared to PRE were solely significantly smaller in the CTRL group (−8.4%; $p < 0.01$, Scheffe post-hoc).

**3.1.2 Maximum repetitions of push-ups (Fig 2B).** The ANOVA revealed no main effect of Group ($F_{(1, 34)} = 1.67$, $p = 0.21$, $\eta^2 = 0.04$), no main Session effect ($F_{(1, 34)} = 0.1$, $p = 0.32$, $\eta^2 = 0.03$), and a significant Group×Session interaction effect ($F_{(1, 34)} = 16.71$, $p < 0.001$, $\eta^2 = 0.54$). No significant difference in maximum repetitions of push-ups was observed between the CTRL and the EXP groups in the PRE-test phase (43.9 ± 15.4 and 49.1 ± 24,6 repetitions in average for the CTRL and the EXP group respectively, $p = 0.57$, Scheffe post-hoc). By contrast, POST-test maximum repetitions of

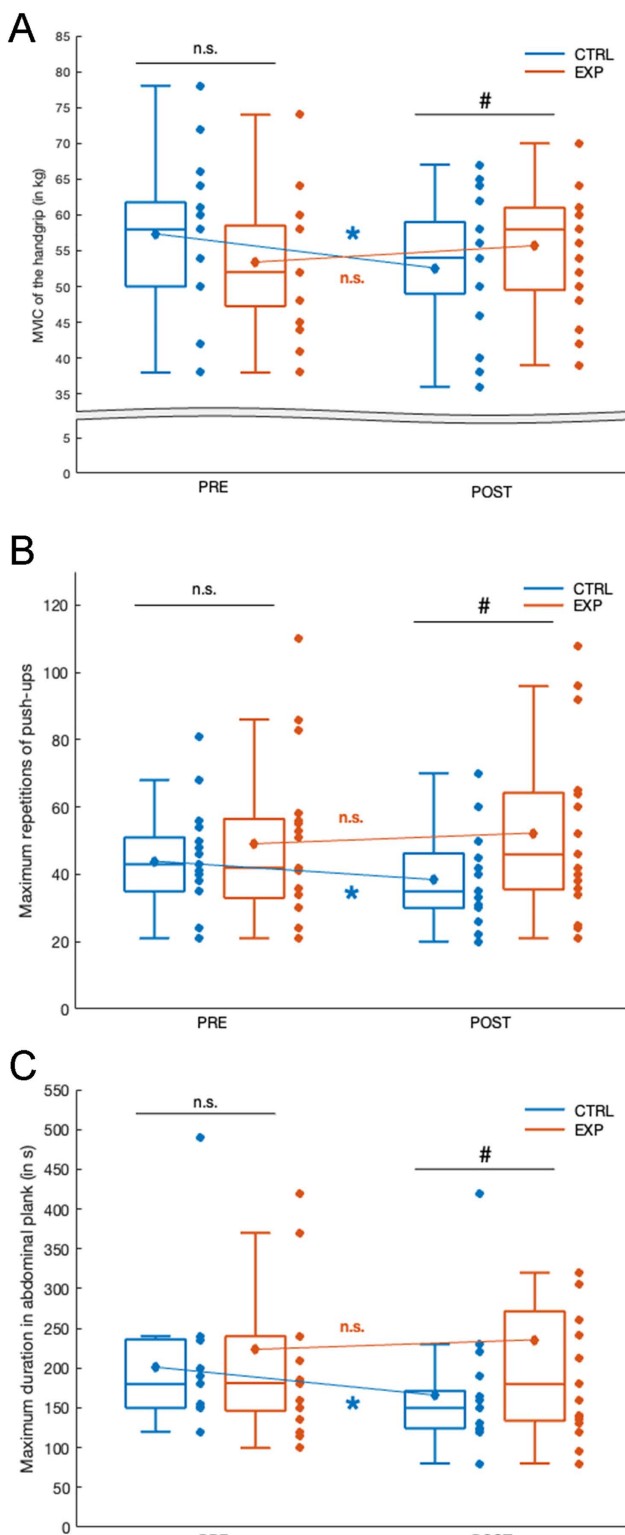

**Fig 2. Muscular performances of Handgrip (A), Push-Ups (B) and Abdominal Plank (C) tests respectively.** The blue blogspots correspond to the CTRL group and the orange ones to the EXP group. Central horizontal bars on boxplots indicate median values, diamonds indicate mean values, boxes represent first through third quartiles, and lower and upper ends correspond respectively to the minimum and maximum values. Scatterplots correspond to the value of each subject in each group. ANOVA effect: n.s. **p.** > 0.05; * **p.** < 0.05; ** **p.** < 0.01; ***. Post Hoc effect: n.s. **p.** > 0.05; # **p.** < 0.05.

push-ups was significantly higher in the EXP group compared to the CTRL group ($52.3 \pm 25.7$ vs. $38.4 \pm 13.2$ repetitions in average for the EXP and the CTRL group respectively, $p = 0.02$, Scheffe post-hoc). The Group×Session interaction was due to the fact that the maximum repetitions of push-ups in the POST-test compared to the PRE-test phase were solely significantly smaller in the CTRL group ($-12.5\%$; $p < 0.01$, Scheffe post-hoc).

### 3.1.3 Maximum duration in abdominal plank (Fig 2C).

The ANOVA revealed no main effect of Group ($F_{(1, 34)} = 1.75$, $p = 0.20$, $\eta^2 = 0.05$), no main Session effect ($F_{(1, 34)} = 2.03$, $p = 0.16$, $\eta^2 = 0.04$), and a significant Group×Session interaction effect ($F_{(1, 34)} = 9.53$, $p < 0.01$, $\eta^2 = 0.42$). No significant difference in the maximum duration in abdominal plank was observed between the CTRL and the EXP groups in the PRE-test phase ($201.2 \pm 83.6$ and $223.6 \pm 134.5$ seconds in average for the CTRL and the EXP group respectively, $p = 0.31$, Scheffe post-hoc). By contrast, POST-test maximum duration in abdominal plank was significantly higher in the EXP group compared to the CTRL group ($235.6 \pm 173.8$ vs. $166.0 \pm 75.3$ seconds in average for the EXP and the CTRL group respectively, $p = 0.01$, Scheffe post-hoc). The Group×Session interaction was due to the fact that the maximum duration in abdominal plank in the POST-test compared to the PRE-test phase were solely significantly smaller in the CTRL group ($-17.5\%$; $p < 0.05$, Scheffe post-hoc).

## 3.2 Borg Rating of Perceived Exertion (RPE)

The ANOVA revealed no main effect of Group ($F_{(1, 34)} = 1.75$, $p = 0.29$, $\eta^2 = 0.04$), a main significant Session effect ($F_{(1, 34)} = 90.14$, $p < 0.001$, $\eta^2 = 0.73$), and no significant Group×Session interaction effect ($F_{(1, 34)} = 0.96$, $p = 0.55$, $\eta^2 = 0.01$) on RPE score. More precisely, average RPE score was $6.85 \pm 1.26$ in PRE, and increased to $8.03 \pm 1.29$ in POST.

## 3.3 Heart Rate (HR) measures

### 3.3.1 HR in supine position ($HR_{SU}$; Fig 3).

The ANOVA revealed no main effect of Group ($F_{(1, 28)} = 2.17$, $p = 0.15$, $\eta^2 = 0.17$), no main Session effect ($F_{(1, 28)} = 0.58$, $p = 0.45$, $\eta^2 = 0.02$), and a significant Group×Session interaction effect ($F_{(1, 28)} = 45.95$, $p < 0.001$, $\eta^2 = 0.64$). No significant difference in the $HR_{SU}$ was observed between the CTRL and the EXP groups in the PRE-test phase ($73.6 \pm 12.4$ and $75.4 \pm 13.9$ bpm in average for the CTRL and the EXP group respectively, $p = 0.70$, Scheffe post-hoc). By contrast, POST-test $HR_{SU}$ was significantly higher in the EXP group compared to the CTRL group ($80.2 \pm 15.2$ vs. $67.6 \pm 10.9$ bpm in average for the EXP and the CTRL group respectively, $p = 0.01$, Scheffe post-hoc).

The training Group×Session interaction was due to the fact that the $HR_{SU}$ in the POST compared to the PRE were solely significantly lower in the CTRL group ($-8.1\%$; $p < 0.001$, Scheffe post-hoc).

### 3.3.2 HR in standing position ($HR_{ST}$; Fig 3).

The ANOVA revealed no main effect of Group ($F_{(1, 28)} = 1.07$, $p = 0.31$, $\eta^2 = 0.03$), no main session effect ($F_{(1, 28)} = 3.14$, $p = 0.09$, $\eta^2 = 0.1$), and a significant Group×Session interaction effect ($F_{(1, 28)} = 79.30$, $p < 0.001$, $\eta^2 = 0.75$). No significant difference in the $HR_{ST}$ was observed between the CTRL and the EXP groups in the PRE-test phase ($92.6 \pm 13.1$ and $92.1 \pm 12.4$ bpm in average for the CTRL and the EXP group respectively, $p = 0.96$, Scheffe post-hoc). By contrast, POST-test $HR_{ST}$ was significantly higher in the EXP group compared to the CTRL group ($98.4 \pm 11.8$ vs. $88.4 \pm 11.3$ bpm in average for the EXP and the CTRL group respectively, $p = 0.03$, Scheffe post-hoc).

The training Group×Session interaction was due to the fact that the $HR_{ST}$ in the POST compared to the PRE were solely significantly lower in the CTRL group ($-4.5\%$; $p < 0.001$, Scheffe post-hoc).

## 3.4 Heart Rate Variability (HRV) in SU

Detailed results of HRV in supine and standing positions (SU and ST, respectively) are presented in Table 2.

### 3.4.1 LF in SU ($LF_{SU}$; Table 2 upper part).

The ANOVA revealed no main effect of Group ($F_{(1, 28)} = 3.08$, $p = 0.09$, $\eta^2 = 0.10$), no main Session effect ($F_{(1, 28)} = 1.22$, $p = 0.27$, $\eta^2 = 0.04$), and a significant Group×Session interaction effect

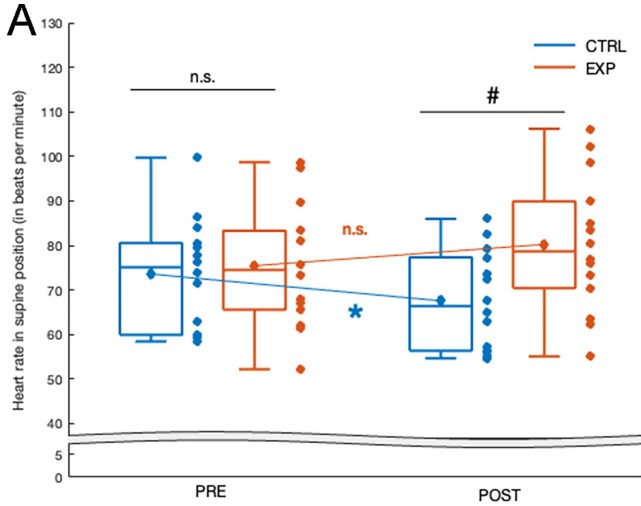

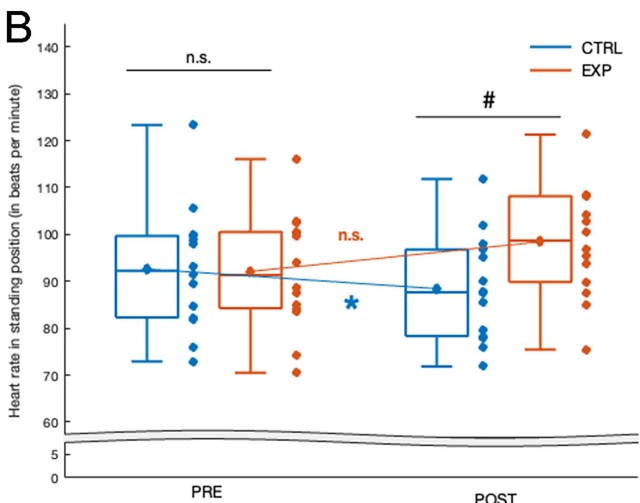

**Fig 3. Heart Rate Measures in Suping Position (A) and Standing Position (B).** The blue blogspots correspond to the CTRL group and the orange ones to the EXP group. Central horizontal bars on boxplots indicate median values, diamonds indicate mean values, boxes represent first through third quartiles, and lower and upper ends correspond respectively to the minimum and maximum values. Scatterplots correspond to the value of each subject in each group. ANOVA effect: n.s. **p.** > 0.05; * **p.** < 0.05; ** **p.** < 0.01; ***. Post Hoc effect: n.s. **p.** > 0.05; # **p.** < 0.05.

(F(1, 28)=6.70, p < 0.05, $\eta^2$ = 0.31). No significant difference in the $LF_{SU}$ was observed between CTRL and EXP in PRE (p = 0.16, Scheffe post-hoc). The significant interaction was due to the fact that $LF_{SU}$ significantly decreased in CTRL and concomitantly increased in EXP between PRE and POST (−27% and +25%, respectively; p < 0.05, Scheffe post-hoc).

**3.4.2 HF in SU ($HF_{SU}$; Table 2 upper part).** The ANOVA revealed no main effect of Group (F(1, 28)=0.06, p = 0.81, $\eta^2$ = 0.002), no main Session effect (F(1, 28)=0.26, p = 0.61, $\eta^2$ = 0.01), and no significant Group×Session interaction effect (F(1, 28)=1.25, p = 0.30, $\eta^2$ = 0.11).

**3.4.3 TP in SU ($TP_{SU}$; Table 2 upper part).** The ANOVA revealed no main effect of Group (F(1, 28)=0.59, p = 0.45, $\eta^2$ = 0.02), a significant session effect (F(1, 28)=18.10, p < 0.001, $\eta^2$ = 0.42), and no significant Group×Session interaction effect (F(1, 28)=1.19, p = 0.29, $\eta^2$ = 0.04).

**Table 2. Average (±SD) of every HRV parameter (column one) when supine (the upper part of the table) and standing (the lower part of the table) in each group (column two) for all the participant (column three) in PRE-tests (column four) and POST-tests (column five). The F and P-values of the ANOVA were reported for the main effect of Group (column six and seven), Session effect (column eight and nine), and Group×Session interaction effect (column ten and eleven). For a sake of clarity, parameters with significant statistical Group×Session interaction were reported with bold characters and the interaction effect analyses are detailed in the text. p.>0,05; * p.<0,005; ** p.<0,01; *** p.<0,001.**

| Parameter (unit) | Group | n | Mean PRE-test (SD) | Mean POST-test (SD) | Group effect | | Session effect | | Interaction effect | |
|---|---|---|---|---|---|---|---|---|---|---|
| Supine (SU) | | | | | *F-Value* | *p-Value* | *F-Value* | *p-Value* | *F-Value* | *p-Value* |
| **LF$_{SU}$ (ms²)** | **CTRL** | **14** | **1148 (1013)** | **840 (697)** | **3.08** | **0.10** | **1.22** | **0.27** | **6.70** | ***** |
| | **EXP** | **14** | **488 (413)** | **611 (549)** | | | | | | |
| HF$_{SU}$ (ms²) | CTRL | 14 | 226 (277) | 578 (561) | 0.06 | 0.81 | 0.26 | 0.61 | 1.25 | 0.30 |
| | EXP | 14 | 567 (1038) | 343 (581) | | | | | | |
| TP$_{SU}$ (ms²) | CTRL | 14 | 1374 (1255) | 1952 (1667) | 0.59 | 0.45 | 18.10 | *** | 1.19 | 0.29 |
| | EXP | 14 | 1056 (1356) | 1399 (1777) | | | | | | |
| **LF/HF$_{SU}$** | **CTRL** | **14** | **10.5 (12.9)** | **2.6 (1.9)** | **2.02** | **0.16** | **3.41** | **0.10** | **5.67** | ***** |
| | **EXP** | **14** | **3.5 (3.7)** | **4.0 (2.8)** | | | | | | |
| Standing (ST) | | | | | | | | | | |
| **LF$_{ST}$ (ms²)** | **CTRL** | **14** | **1157 (1568)** | **868 (1449)** | **0.9** | **0.76** | **0.42** | **0.52** | **12.80** | ****** |
| | **EXP** | **14** | **774 (838)** | **974 (920)** | | | | | | |
| HF$_{ST}$ (ms²) | CTRL | 14 | 262 (547) | 125 (171) | 0.64 | 0.43 | 0.31 | 0.58 | 1.24 | 0.27 |
| | EXP | 14 | 115 (174) | 112 (140) | | | | | | |
| **TP$_{ST}$ (ms²)** | **CTRL** | **14** | **1420 (2096)** | **1033 (1603)** | **0.16** | **0.70** | **0.54** | **0.46** | **8.70** | ***** |
| | **EXP** | **14** | **890 (953)** | **1122 (1034)** | | | | | | |
| **LF/HF$_{ST}$** | **CTRL** | **14** | **18.1 (30.7)** | **12.5 (19.9)** | **0.03** | **0.90** | **1.01** | **0.32** | **5.02** | ***** |
| | **EXP** | **14** | **15.8 (21.3)** | **17.7 (20.2)** | | | | | | |

**3.4.4 LF/HF ratio in SU (LF/HF$_{SU}$; Table 2 upper part).** The ANOVA revealed no main effect of Group (F(1, 28)=2.02, p=0.16, η²=0.07), no significant Session effect (F(1, 28)=3.41, p=0.09, η²=0.14), and a significant Group×Session interaction effect (F(1, 28)=5.67, p<0.05, η²=0.27). No significant difference in the LF/HF$_{SU}$ was observed between the CTRL and the EXP groups in PRE (p=0.10, Scheffe post-hoc). The significant Group×Session interaction could be explained by the fact that the LF/HF$_{SU}$ in POST compared to PRE decreased in the CTRL while it increased in the EXP group (−75% and +14% in average respectively; p<0.05, Scheffe post-hoc).

### 3.5 HRV in ST

**3.5.1 LF in ST (LF$_{ST}$; Table 2 lower part).** The ANOVA revealed no main effect of Group (F(1, 28)=0.90, p=0.76, η²=0.3), no main Session effect (F(1, 28)=0.42, p=0.52, η²=0.03), and a significant Group×Session interaction effect (F(1, 28)=12.80, p<0.01, η²=0.42). No significant difference in the LF$_{ST}$ was observed between the CTRL and the EXP groups in PRE (p=0.16, Scheffe post-hoc). The significant Group×Session interaction could be explained by the fact that the LF$_{ST}$ in POST compared to PRE decreased in the CTRL while it increased in the EXP group (−25% and +26% in average respectively; p<0.05, Scheffe post-hoc).

**3.5.2 HF in ST (HF$_{ST}$; Table 2 lower part).** The ANOVA revealed no main effect of Group (F(1, 28)=0.64, p=0.43, η²=0.002), no main Session effect (F(1, 28)=0.31, p=0.58, η²=0.001), and no significant Group×Session interaction effect (F(1, 28)=1.24, p=0.25, η²=0.004).

**3.5.3 TP in ST (TP$_{ST}$; Table 2 lower part).** The ANOVA revealed no main effect of Group (F(1, 28)=0.16, p=0.70, η²=0.06), no main Session effect (F(1, 28)=0.54, p=0.46, η²=0.02), and a significant Group×Session interaction effect

($F(1, 28)=8.70$, $p<0.05$, $\eta^2=0.35$). No significant difference in the $TP_{ST}$ was observed between the CTRL and the EXP groups in PRE ($p=0.44$, Scheffe post-hoc). The significant Group×Session interaction could be explained by the fact that the $TP_{ST}$ in POST compared to PRE decreased in the CTRL while it increased in the EXP group (−27% and +26% in average respectively; $p<0.05$, Scheffe post-hoc).

**3.5.4 LF/HF ratio in ST ($LF/HF_{ST}$; Table 2 lower part).** The ANOVA revealed no main effect of Group ($F(1, 28)=0.03$, $p=0.89$, $\eta^2=0.06$), no significant Session effect ($F(1, 28)=1.01$, $p=0.32$, $\eta^2=0.02$), and a significant Group×Session interaction effect ($F(1, 28)=5.02$, $p<0.05$, $\eta^2=0.35$). No significant difference in the $LF/HF_{ST}$ was observed between the CTRL and the EXP groups in PRE ($p=0.98$, Scheffe post-hoc). The significant Group×Session interaction could be explained by the fact that the $LF/HF_{ST}$ in POST compared to PRE decreased in the CTRL while it increased in the EXP group (−31% and +12% in average respectively; $p<0.05$, Scheffe post-hoc).

## 4. Discussion

The primary aim of this study was to investigate whether incorporating mental imagery and conscious breathing exercises into a conventional and standardized warm-up routine could optimize cardiac autonomic modulation, muscular strength, and endurance performance in firefighters. To address this question, two groups performed the same physical warm-up, with the experimental group also engaging in psychophysiological techniques during the exercises. Cardiac autonomic activity was assessed through HR and HRV indices, while maximal voluntary isometric handgrip contraction, maximal push-ups, and abdominal plank duration were used to evaluate muscle strength and endurance. Our findings indicate that integrating mental imagery and controlled breathing elicited greater sympathetic activation and higher HR responses compared with the control warm-up alone. This modulation of the ANS was accompanied by attenuated declines in repeated maximal performance, suggesting that psychophysiological practices helped mitigate both central and peripheral fatigue. Taken together, these results support the integration of targeted psychophysiological strategies into operational warm-up routines to enhance readiness and performance in tactical populations such as firefighters.

### 4.1 Perception of exertion and fatigue when repeating twice maximum physical performances

Before anything else, it is important to note that the present results do not show any difference in the psychological perception of exertion between the two groups. Indeed, RPE scores were similar at each stage between the CTRL and the EXP groups. Consequently, the discussions of the physiological results below on the effects of the addition of mental imagery and breathing techniques are always conducted at similar RPE levels between the two groups.

Subsequently, from a behavioral performance perspective, the present results showed, on one hand, that maximal performance in the POST-tests was significantly reduced in the CTRL group for the MVIC hand grip test (−8.40%), the maximum push-ups (−12.5%), and the maximum duration of the abdominal plank (−17.5%). These observed decreases in muscular performance should not be attributed to the warm-up itself but rather to the physiological cost of the testing protocol, which involved successive maximal and submaximal efforts (i.e., handgrip MVIC represents a maximal effort, push-ups entails submaximal efforts until exhaustion, and the abdominal plank-test involves a sustained submaximal isometric contraction to exhaustion) known to induce both central and peripheral fatigue [43,44]. In this context, the warm-up (particularly when supplemented with mental imagery and controlled breathing) does not prevent fatigue but modulates its magnitude and physiological impact. Accordingly, the EXP group maintained higher performance levels than the CTRL group, suggesting that psychophysiological practices helped mitigate fatigue-effects, likely through enhanced autonomic regulation and corticospinal excitability.

## 4.2 The addition of mental imagery practices and breathing techniques to conventional warm-up optimize physical performance when repeating twice maximum physical tests

In contrast, when imagery practices and controlled breathing techniques were integrated into the warm-up routine (EXP group), participants demonstrated preserved performance between PRE and POST-assessments, as evidenced by results in MVIC handgrip strength (+4.3%), maximum number of push-ups (+6.5%), and maximal abdominal plank duration (+5.4%). Post hoc analyses showed that, in the post-test phase, the EXP group's performance in the physical tests was significantly higher than that of the CTRL group, whereas no such difference was observed during the pre-test phase. These outcomes substantiate the initial component of our hypothesis, indicating that, under conditions of repeated maximal physical testing, firefighters who incorporate both neuromuscular warm-up and psyching-up strategies such as mental imagery and respiratory control are less susceptible to performance decrements.

Firstly, such results are in line with the literature dealing with mental imagery practices and more particularly the recent works of Rumeau et al. [45,46] which reported among athletes that addition of motor imagery emphasized the effects of a standardized warm-up on sprint running performance, reaction time, strength repeated effort ability. More widely, mental imagery is well known to increase strength performance [47–49] and strength endurance [50] which are the main athletic qualities involved in the tests used in the present work as well as one of the main athletic qualities involved in firefighters exams and on duty [5,9]. Moreover, the maintaining muscular performances highlighted by our results may also be explained by the fact that motor imagery was able to compensate neuromuscular fatigue when maximal and submaximal muscular contractions were repeated [27].

In these previous work, mental imagery protocols have typically been implemented over extended periods -ranging from several hours to multiple weeks- rendering their practical application on a daily basis infeasible for on-duty firefighters due to operational constraints. Using a novel and pragmatic approach, the present study provides evidence that a brief intervention consisting of only eight minutes of mental imagery, cumulatively distributed across the 13-minute protocol and systematically performed alongside physical exercises, sometimes in combination with controlled breathing, is sufficient to enhance muscular performance during repeated maximal strength tasks in firefighters. This conclusion is further supported by the findings of Grosprêtre et al. [51] who highlighted that a single session of mental imagery was sufficient to induce presynaptic inter-neuronal plasticity, an effect known to enhance muscular performance. Therefore, when appropriately tailored to the operational constraints of on-duty firefighters, mental imagery emerges as a viable and effective strategy to be integrated into both training regimens and routine professional activities, with the aim of optimizing repeated strength performance.

Secondly, the present findings are consistent with existing literature on breathing techniques, which has shown, particularly among college students and athletes, that certain practices, such as yogic breathing modules or *pranayama* (a technique involving phases of inhalation, breath retention, and exhalation), can enhance peripheral oxygen saturation [52], muscular strength, and muscular endurance [53]. Furthermore, depending on the specific inhalation-to-exhalation ratio employed, these techniques may significantly influence not only muscular performance but also heart rate variability (HRV) and heart rate (HR), thereby improving ANS adaptability to intense psychological and physiological demands [52]: a particularly relevant benefit given the high-stress, high-intensity nature of firefighters' operational duties [1].

## 4.3 The addition of mental imagery practices and breathing techniques to conventional warm-up routine promotes sympathetic activation, reduces vagal work and optimizes the heart rate in both supine and standing positions

Our HRV-related results revealed significant interaction effects on low-frequency components (LFs). Specifically, post-hoc analyses showed a substantial increase in LF power in the EXP group (+25.5% on average), commonly associated with sympathetic modulation [54], whereas a significant decrease was observed in the CTRL group (−26% on average), across

both supine and standing positions. These findings were further supported by the LF/HF ratio, a recognized marker of sympathovagal balance [54], which similarly exhibited significant interaction effects. Post-hoc tests indicated that the LF/HF ratio increased in the EXP group (+13% on average) but decreased markedly in the CTRL group (−53% on average). Concurrently, heart rate (HR) data revealed significant interaction effects as well, with the CTRL group showing a lower HR (−6.3% on average) across both body positions.

Taken together, these HRV and HR findings robustly support the second part of our hypothesis: firefighters who completed a standardized warm-up routine incorporating both mental imagery and breathing exercises exhibited heightened sympathetic activation and elevated HR POST-intervention, compared to those who performed only conventional warm-up. These autonomic responses align with existing literature on the conscious regulation of respiration and the physiological effects of mental imagery techniques. Indeed, the scientific literature reported a causal link between the breathing frequencies and the dominant activation of the parasympathetic or sympathetic system [55,56]. For example, cardiac coherence techniques based on 6 breaths per minute with an equal inhalation/exhalation ratio of 5 seconds are known to increase parasympathetic tone [57,58]. Noteworthy is that most studies focused on the parasympathetic activation [52,59] in order to increase the state of calm [28], to reduce the perception of pain [60] or the anxiety [57]. Conversely and in an original way, the present study highlight that conscious regulation of breathing rate with full long inspiration followed by sharp and short expiration as well as short apnea [61] can increase sympathetic tone and HR frequencies for the purpose of optimizing a warm-up phase. It is consistent with previous studies which showed that sympathetic tone and HR frequencies may be increased with (i) a ratio favors inspiratory over expiratory duration [62], and (ii) apneas generating an oxygen debt [63], and in agreement with the fact that neural activities are able to increase HR just before and at the beginning of real as well as purely mental physical exercises [22], thus promoting faster respiratory and cardiac modulation in order to optimize the coming physical exercise.

In summary, increased sympathetic activation, reflected by elevated HR and shifts in HRV indices, likely contributes to muscular performance maintenance by improving cardiac output and muscle perfusion, and increasing neuromuscular excitability. In the EXP group, this autonomic facilitation coincided with smaller declines in repeated maximal tasks, supporting a mechanistic link between sympathetic priming and sustained force and endurance performance.

## 5. Conclusion

This study is among the first to investigate the impact of incorporating psyching-up techniques into standard professional routines on both cardiac activity and physical performance in firefighters. Our findings clearly demonstrate that adding mental imagery and controlled breathing exercises to a conventional warm-up enhances sympathetic activation, elevates heart rate, and ultimately optimizes subsequent physical performance. Given the repeated high-intensity physical demands and associated fatigue that typify firefighting missions [5], we advocate for the systematic integration of such targeted psyching-up strategies into existing operational protocols. These interventions offer a cost-effective and time-efficient means of optimizing autonomic and cardiovascular readiness prior to deployment, thereby enhancing the performance capacity of first responders across French emergency services.

## 6. Study limits

This study has several limitations. Only 28 of the 34 participants were equipped with HRV sensors due to equipment availability, although all completed the physical tests. HRV indices – particularly LF and HF parameters – showed substantial inter-individual variability, likely reflecting differences in imagery ability, breathing control, and baseline autonomic regulation. Future studies should therefore examine individual characteristics more closely, especially imagery ability

and responsiveness to imagery practice, which requires systematic evaluation of participants' capacity to generate and perform motor imagery. Motor imagery could not be directly monitored during exercise execution in the absence of neuro-physiological measures; laboratory-based approaches, such as fMRI, could help address this limitation. Furthermore, the 15-minute HRV test protocol necessitated a second warm-up to ensure participant safety before the subsequent muscular tests. This may have modulated the magnitude of observed effects, although the procedure was strictly identical across groups to prevent comparison bias. Finally, because the protocol involved repeated maximal and submaximal efforts to exhaustion, the observed performance decrements reflect the physiological cost of the testing sequence rather than a direct effect of the warm-up; thus, findings should be interpreted as relative group differences. Future research should aim to implement comprehensive HRV monitoring, evaluate long-term adaptations, and explore the operational transfer of such psychophysiological routines in first responder contexts.

## 7. Future prospects

While our protocol is ecologically valid, further refinement could be achieved by incorporating more targeted exercises. A parallel study utilizing a Multiple Single Case Experimental Design (31) has already explored a similar intervention, simulating operational conditions within a fire truck. In this study, a combination of psyching-up techniques was introduced to firefighters during simulated interventions over a 24-hour on-duty period. Future directions will focus on gathering feedback from firefighters, as well as input from team leaders, regarding the feasibility and practical application of these psyching-up practices. Additionally, assessing the real costs of training and implementing these techniques after one full experimental year in real-world settings with partner fire and rescue centers will be crucial.

## 8. Appendices

Annexe 1: Eligibility criteria for participating firefighters.

1. aged between 22 and 50 with at least 5 years' experience as a firefighter;

2. required to have a physical check-up by the local firefighter's healthcare institution, and received medical clearance to perform firefighting activities;

3. being affiliated to the French social security system and having health insurance;

4. not having had any musculoskeletal disorders or recent musculoskeletal injuries;

5. not having suffered from post-traumatic stress disorder;

6. no neurological history;

7. not following acute or chronic drug treatment;

8. having a normal or corrected-to-normal hearing;

9. not consuming neuro-stimulants or energy drinks 24 hours before and during the experiment;

10. having a minimum of 3 consecutive days off without professional work;

11. not undergoing sports training during the 72-hours before the experiment.

## Supporting information

**S1 Dataset. Complete raw and aggregated data for muscular performance and heart rate variability outcomes.**
This Excel file contains the full dataset used in the analyses, structured into three sheets: "Muscular Performances": individual pre- and post-intervention scores for maximal handgrip force (HGF, kg), push-ups completed (PU, units), plank

hold duration (PL, seconds), and rating of perceived exertion (RPE). The sheet includes group allocation and descriptive statistics (means and standard deviations) for both the intervention and control groups. "HRV SU": raw pre- and post-intervention heart rate variability (HRV) data recorded in the supine position, including time-domain and frequency-domain metrics (e.g., RMSSD, LF, HF, total power, LF/HF ratio), along with descriptive statistics. "HRV ST": raw pre- and post-intervention HRV data recorded in the standing position, including the same HRV metrics as above, with individual values and group-level descriptive statistics. All variables are presented here in their original units exactly as collected during the experimental procedures.
(XLSX)

## Author contributions

**Conceptualization:** Jean Philippe Biéchy, Camille Charissou, Thomas Skrysinski, Sylvain Gobert, David Amarantini, Lilian Fautrelle.

**Data curation:** Jean Philippe Biéchy, David Amarantini, Lilian Fautrelle.

**Formal analysis:** David Amarantini, Lilian Fautrelle.

**Funding acquisition:** Thomas Skrysinski, Lilian Fautrelle.

**Investigation:** Jean Philippe Biéchy, Camille Charissou, Candice Groléas, Thomas Skrysinski, David Amarantini, Lilian Fautrelle.

**Methodology:** David Amarantini, Lilian Fautrelle.

**Project administration:** Thomas Skrysinski, Lilian Fautrelle.

**Resources:** Candice Groléas, Thomas Skrysinski, Sylvain Gobert, Lilian Fautrelle.

**Software:** David Amarantini, Lilian Fautrelle.

**Supervision:** Sylvain Gobert, David Amarantini, Lilian Fautrelle.

**Validation:** Camille Charissou, Sylvain Gobert, David Amarantini, Lilian Fautrelle.

**Visualization:** Jean Philippe Biéchy, Camille Charissou, Candice Groléas, Sylvain Gobert, David Amarantini, Lilian Fautrelle.

**Writing – original draft:** Jean Philippe Biéchy, Camille Charissou, David Amarantini, Lilian Fautrelle.

**Writing – review & editing:** Jean Philippe Biéchy, Camille Charissou, Candice Groléas, Thomas Skrysinski, Sylvain Gobert, David Amarantini, Lilian Fautrelle.

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
