## [Decision Letter · Decision Letter 0]

9 Jul 2025

Dear Dr. Amarantini,

Thank you for submitting your manuscript to PLOS ONE. After careful consideration, we feel that it has merit but does not fully meet PLOS ONE’s publication criteria as it currently stands. Therefore, we invite you to submit a revised version of the manuscript that addresses the points raised during the review process.

**ACADEMIC EDITOR:**

Dear Authors,

one expert in the field reviewed your manuscript detecting several major issues you should consider during the revision process. 

We look forward to receiving your revised manuscript.

Kind regards,

Emiliano Cè, Ph.D.

Academic Editor

PLOS ONE

Journal Requirements:

1. Please ensure that your manuscript meets PLOS ONE's style requirements, including those for file naming. The PLOS ONE style templates can be found at https://journals.plos.org/plosone/s/file?id=wjVg/PLOSOne_formatting_sample_main_body.pdf and https://journals.plos.org/plosone/s/file?id=ba62/PLOSOne_formatting_sample_title_authors_affiliations.pdf.

2. In the online submission form, you indicated that [he data underlying the results presented in the study are available from David Amarantini, Laboratoire ToNIC, david.amarantini@inserm.fr].

Reviewers' comments:

Reviewer's Responses to Questions

**Comments to the Author**

1. Is the manuscript technically sound, and do the data support the conclusions?

Reviewer #1: Yes

2. Has the statistical analysis been performed appropriately and rigorously?

Reviewer #1: Yes

3. Have the authors made all data underlying the findings in their manuscript fully available?

Reviewer #1: Yes

4. Is the manuscript presented in an intelligible fashion and written in standard English?

Reviewer #1: Yes

Reviewer #1: The present study investigated the effects of mental imagery and breathing added to a conventional warm-up on the sympathetic activation and muscle force and endurance. Some major concerns should be addressed to clarify the rationale (introduction), to justify the methods (2 repeated warm-up sessions for example), to clarify the statistics (e.g. huge variability) and to make the discussion clear. Please see below.

Title and Abstract.

The title is clear and the abstract almost adequately describe the study. The methods (e.g., the 2 experimental groups? the muscle group?) and results (e.g., physical performances?) should be developed for clarity. The last sentence is very general and not really helpful. Please consider rephrasing and giving a clear conclusion related to the present findings.

Introduction.

L51. high level of cardiac stress is too restrictive.

L53. What does 'job performance' mean?

L67-L75. The message behind this paragraph is unclear. What is the link with the preceding and following paragraphs.

L81-82. It is unclear why activating the sympathetic system may be beneficial. The optimisation of warm-up routines is easier to understand. It can be achieved while lowering fatigue associated with the different warm-up exercises and while optimizing some physiological systems (for example the sympathetic). But the usefulness of the sympathetic system should be made clearer.

L89-95. Authors seem to indicate these effects are related to the autonomic nervous system. Please explain or clarify. Generally speaking, the effect of motor imagery on the autonomic nervous system is unclear. Similarly, the link between this autonomic system and strength is unclear.

L108-112. The hypothesis should also be clarified. Why less performance decline? The exercise considered should be defined here.

Methods.

L132-135. What is the rationale for considering 2 experimental groups and not considering a cross-over design? Please justify the sample size.

L146. How were volunteers during the 30-min rest (seated, lying...)?

L147. Why only 28 with HRV measurements? What were not involved volunteers doing during these 15 minutes?

L149. Figure 1 indicates 15 minutes for HRV test (also L154).

L154-155. HRV was evaluated after a single warm-up while muscle force/endurance were evaluated after 2 warm-up protocols. Have authors tested the potential bias? Were muscle properties similarly affected after a single or double warm-up procedure? Same question for HRV.

L166, L170 and table 1. It is not table A1 but table 1. It is unclear if participants are doing mental imagery during or following the warm-up exercice. How did authors controlled the intensity of each exercices between CTRL and EXP. Did authors control the breathing and mental imagery? and how?

L180. How many contractions? possible rest duration? Rest between this test and push-ups? Moreover, push-up is a form of dynamic plank. What is the influence of this pre-exercice. It can serve as a fatiguing exercice or as an additional warm-up exercice.

L199. 6-min rest was observed between push-up and plank. This long rest could reduce the warm-up effects. Was the effect of this delay tested?

L210. Conducting warm-up + mental imagery + controlled breathing could increase the (warm-up) perceived exertion. Why was RPE not evaluated following warm-up.

L223. How was respiratory rate measured?

Results.

Please be consistant while writing pre/post or PRE/POST.

Results and Figures. Figure should be modified to better show the * related to the 'performance' decrease in the control group. Also, was performance similar between groups post-warm-up? In the text, values during pre-tests are useless and redundant with figure 2.

L290. Use '.' and not ',' for values.

Table 2. One can question about the variability obtained for LF and HF. How can authors explain this huge inter-individual and inter-group variation? How can authors explain the lack of difference in pre-values between groups. These results are questionable.

Discussion.

L387-397. The aim of the study is different than the aim presented in introduction. Similarly authors indicate comparing percentage change. It was not done. In addition, some new terms are used (dynamization). Please be consistent throughout the manuscript.

L399-415. This part is unclear and is too briefly discussed. Authors indicate MVIC decrease is due to fatigue. How it it possible? A warm-up should limit fatigue. What does authors mean with repeating maximal and submaximal contractions. This repetition induce fatigue? So it means that authors have not measured any warm-up effects. Tests per se bias the results?

L428. I still can't understand the hypothesis about performance decrements. A warm-up is not conducted to produce fatigue.

L438-441. Same comment than for part 4.1.

L446. Where do these 8minutes come from? Is it just from mental imagery or the combination of breathing and imagery?

Part 4.3. Can authors make the link between sympathetic activation and performance (cardiac, muscle or endurance). The likely effects and applications are not so clear.

L517-528. I disagree with the conclusion. imagery and breathing do not improves performance. It is not shown by statistics.

Part 6. Some limitations could be added.

**Do you want your identity to be public for this peer review?** For information about this choice, including consent withdrawal, please see our Privacy Policy

Reviewer #1: No

---

## [Author Response · Author response to Decision Letter 1]

14 Oct 2025

Title and Abstract.

The title is clear and the abstract almost adequately describe the study. The methods (e.g., the 2 experimental groups? the muscle group?) and results (e.g., physical performances?) should be developed for clarity. The last sentence is very general and not really helpful. Please consider rephrasing and giving a clear conclusion related to the present findings.

We have modified the abstract accordingly as follows (highlighted in yellow) to address these comments:

“The capacity of firefighters to consistently mobilize their full physical potential during duty shifts is of paramount importance. One promising approach to achieving this goal involves developing operational protocols that effectively activate the sympathetic nervous system and vagal regulation prior to each professional action. This study investigated the effects of combining mental imagery and conscious breathing exercises with a conventional and standard warm-up on cardiac modulations, muscular strength and endurance performance in 34 firefighters randomly assigned to one of two groups: a control group (n=17), which performed only the conventional and standard warm-up, and an experimental group (n=17), which performed the standard warm-up in combination with mental imagery and breathing exercises. The results showed that incorporating such psychophysiological techniques into a conventional warm-up during repeated physical efforts optimizes sympathetic spectral power, heart rate responses, and physical performances, including maximal voluntary isometric handgrip contraction, maximum push-ups, and the maximal abdominal plank duration. Altogether, these findings indicate that brief mental imagery and controlled breathing exercises incorporated into standard warm-up routines effectively preserve muscular performance under repeated exertion while enhancing sympathetic activation, offering a practical and immediately applicable strategy to optimize firefighters’ operational readiness.”

Introduction.

L51. high level of cardiac stress is too restrictive.

We have modified accordingly as follows (highlighted in yellow) to address these comments:

“Firefighters are required to operate in hostile environments that involve multiple highly demanding physical tasks combined with highly engaging emotional contexts, both of which elicit substantial cardiovascular strain—characterized by pronounced elevations in heart rate—while rescuing victims and combating fires, often at the risk of their own lives (Gendron et al., 2019).”

L53. What does 'job performance' mean?

We have revised this sentence to specify the operational components of job performance in the firefighting context, namely the ability to effectively and safely perform physically demanding firefighting tasks (e.g., victim rescue, hose handling, ladder climbing, equipment carrying), which are commonly used as validated occupational performance indicators in the scientific literature.

We have modified accordingly as follows (highlighted in yellow) to address these comments:

“A high level of physical fitness, muscular endurance, and muscular strength has been associated with improved performance in operational firefighting tasks—such as victim rescue, hose handling, ladder climbing, and equipment carrying—as well as with a reduced risk of injury (Johnson et al., 2022; Michaelides et al., 2008; Mattila et al., 2007).”

L67-L75. The message behind this paragraph is unclear. What is the link with the preceding and following paragraphs.

We agree that the original paragraph -L67-75- on the neurophysiological definitions of muscular strength and endurance was too artificial and insufficiently connected to the surrounding sections. To address this, we have revised the text to explicitly link these definitions to the mechanisms through which fatigue impacts firefighters’ physical performance capacity, and to the rationale for exploring countermeasures that target both peripheral and central determinants of performance. This clarification aims now to create a smoother logical transition from the description of operational fatigue factors to the presentation of our psychophysiological intervention strategy.

We have modified accordingly as follows (highlighted in yellow) to address these comments:

“From a neurophysiological perspective, muscular strength refers to the capacity to generate force during a single, brief, maximal voluntary contraction of a muscle at rest (Lewis & Haller, 1991), whereas dynamic and isometric muscular endurance can be defined as the ability to sustain or repeat muscular effort across multiple dynamic movements or during prolonged postural maintenance (Larsson & Karlsson, 1978). Importantly, these physical capacities are governed not only by peripheral musculoskeletal factors but are also strongly modulated by neural mechanisms, including the central nervous system (CNS) and the autonomic nervous system (ANS), which regulates heart rate (HR) and heart rate variability (HRV). This neurophysiological regulation plays a crucial role in determining how fatigue develops and how performance is maintained under operational constraints (Schmitt et al., 2013).”

L81-82. It is unclear why activating the sympathetic system may be beneficial. The optimization of warm-up routines is easier to understand. It can be achieved while lowering fatigue associated with the different warm-up exercises and while optimizing some physiological systems (for example the sympathetic). But the usefulness of the sympathetic system should be made clearer.

Activating the sympathetic nervous system before exertion primes cardiovascular and neuromuscular responses, thereby may optimize warm-up efficiency and subsequent performance (Tran et al., 2001; Csala et al., 2021).

We have added accordingly the following sentences (highlighted in yellow) to address these comments:

“From a neurophysiological perspective, muscular strength refers to the capacity to generate force during a single, brief, maximal voluntary contraction of a muscle at rest (Lewis & Haller, 1991), whereas dynamic and isometric muscular endurance can be defined as the ability to sustain or repeat muscular effort across multiple dynamic movements or during prolonged postural maintenance (Larsson & Karlsson, 1978). Importantly, these physical capacities are governed not only by peripheral musculoskeletal factors but are also strongly modulated by neural mechanisms, including the central nervous system (CNS) and the autonomic nervous system (ANS), which regulates heart rate (HR) and heart rate variability (HRV). This neurophysiological regulation plays a crucial role in determining how fatigue develops and how performance is maintained under operational constraints (Schmitt et al., 2013). In this context, activating the sympathetic nervous system prior to strenuous effort enhances cardiovascular readiness and facilitates greater muscle recruitment, thereby improving strength and endurance performance (Tran et al., 2001; Csala et al., 2021). Such pre-activation primes the body to meet acute energetic demands by increasing heart rate, blood flow, and neuromuscular excitability, thereby potentially optimizing warm-up efficacy.

Within this framework, and without modifying firefighters' work schedules (in France, in the present study), two potential countermeasures could help prevent […]”

L89-95. Authors seem to indicate these effects are related to the autonomic nervous system. Please explain or clarify. Generally speaking, the effect of motor imagery on the autonomic nervous system is unclear. Similarly, the link between this autonomic system and strength is unclear.

“effect of motor imagery on the autonomic nervous system”: Motor imagery can influence the autonomic nervous system by modulating heart rate and heart rate variability, reflecting shifts in sympathovagal balance. Such effects suggest that imagery not only primes corticospinal pathways but may also contribute to autonomic adjustments that facilitate physical readiness (Decety, 1996; Grosprêtre et al., 2016; Bouguetoch et al., 2021).

“link between this autonomic system and strength”: Sympathetic nervous system activation enhances muscular strength and endurance by increasing heart rate, blood pressure, and muscle blood flow, thereby facilitating oxygen delivery and metabolite clearance. In parallel, it augments neuromuscular excitability and motor unit recruitment, mechanisms that together support improved force production and resistance to fatigue (Lombardi et al., 1996; Tran et al., 2001; Csala et al., 2021; Kazuo et al. 2000; Pinto et al. 2017).

In line with this comment, we have now clarified the known effects of emotional visualization and motor imagery on the autonomic nervous system and on strength levels. We have added accordingly the following sentences (highlighted in yellow) to address these comments:

“Firstly, certain mental training techniques, such as motor imagery (Bouguetoch et al., 2021) and emotional visualization (Coombes et al., 2009), have been identified as potential methods to enhance corticospinal excitability, thereby promoting muscle strength and endurance. Although the primary and well-established effects of motor imagery are neural and centrally mediated, several studies also suggested that it can influence autonomic responses, as evidenced by changes in heart rate and heart rate variability (Oishi et al. 2000; Peixeito Pinto et al. 2017 ).”

Added references:

• Peixoto Pinto T, Mello Russo Ramos M, Lemos T, Domingues Vargas C, Imbiriba LA. Is heart rate variability affected by distinct motor imagery strategies? Physiol Behav. 2017 Aug 1;177:189-195. doi: 10.1016/j.physbeh.2017.05.004

• Kazuo Oishi, Tatsuya Kasai, Takashi Maeshima, Autonomic Response Specificity during Motor Imagery, Journal of PHYSIOLOGICAL ANTHROPOLOGY and Applied Human Science, 2000, Volume 19, Issue 6, Pages 255-261

L108-112. The hypothesis should also be clarified. Why less performance decline? The exercise considered should be defined here.

“Why less performance decline”: When two maximal efforts are performed in succession (particularly those involving strength endurance to failure, such as the maximum number of push-ups or the maximal duration of the abdominal plank used in this study) the second performance is generally reduced due to both peripheral and mental fatigue induced by the first effort. This pattern was also observed in our previous work (Biéchy et al. 2021). Therefore, we hypothesized that, during the second bout of maximal testing, the experimental group (benefiting from the addition of psychophysiological practices to their usual warm-up) would experience a smaller decline in performance compared to the control group without such practices.

This situation of repeated high-intensity efforts is also of practical relevance, as it closely reflects the real working conditions of firefighters, who often perform successive interventions during their duty shifts with little or no opportunity for recovery.

We have modified accordingly the following sentences (highlighted in yellow) to address these comments:

“We hypothesized that firefighters performing a standard warm-up supplemented with mental imagery and breathing exercises would exhibit a smaller performance decline —or even performance enhancement— during repeated maximal efforts, including maximal voluntary isometric handgrip contraction, maximum number of push-ups, and maximal abdominal plank duration. These practices were expected to help counteract both central and peripheral fatigue while enhancing corticospinal excitability, cardiovascular readiness and sympathetic activation, compared with controls who performed only the conventional warm-up without psychophysiological techniques.”

Methods.

L132-135. What is the rationale for considering 2 experimental groups and not considering a cross-over design? Please justify the sample size.

“2 experimental groups and not considering a cross-over design:” Regarding the choice of design. We opted for two parallel groups rather than a cross-over because the intervention (mental imagery + controlled breathing integrated into the warm-up routine) is likely to produce carryover effects on both autonomic state (lingering sympathovagal modulation) and behavior (learning/expectancy and psyching-up skills), which a short washout would not reliably eliminate. In addition, our testing sequence already involves repeated maximal efforts (handgrip MVIC, maximal push-ups, maximal abdominal plank) that induce central and peripheral fatigue. In a cross-over this fatigue and recovery trajectory would confound period/sequence effects and require long washouts incompatible with firefighters’ duty schedules. And finally, the parallel-group design maximizes ecological validity, mirroring how such psychophysiological routines would be implemented operationally.

“Please justify the sample size”: A priori, we powered the Group × Session interaction of a mixed ANOVA on our primary outcomes (maximal performances and HR/HRV indices), targeting a medium effect size (f = 0.25–0.30), α = .05, power (1–β) = .80, correlation among repeated measures ≈ .50, non-sphericity = 1.0. This effect size was chosen because it corresponds to the magnitude generally observed in this type of psychophysiological intervention, and particularly in our previous studies (Biéchy et al., 2021; Jamous et al., 2024). Using G*Power (RM-ANOVA, between-factor), this yielded a required total sample of ≈ 26–32 participants. Anticipating ~10–15% attrition and occasional unusable HRV files, we planned n = 34 and achieved this target, with random allocation of 17 participants per group, of whom 28 were equipped with a heart rate monitor, which still meets our targeted statistical power.

We have added accordingly the following sentences (highlighted in yellow) to address these comments:

“Based on G*Power calculations (f = 0.25–0.30), α = .05, and power = .80, a total of 28–32 participants was required, consistent with prior psychophysiological studies (Biéchy et al., 2021; Jamous et al., 2024). We therefore recruited 34 firefighters (age: 34.9 ± 6.9 years; height: 178 ± 4.2 cm; weight: 77.4 ± 5.8 kg; body mass index: 24.6 ± 1.1 kg/m2) volunteered for the experiment.”

L146. How were volunteers during the 30-min rest (seated, lying...)?

During the 30-min rest period, volunteers remained in a standing posture on both feet. All parameters potentially influencing the autonomic nervous system were strictly controlled and standardized across groups, including hydration status, restroom use, and the maintenance of this identical passive standing position.

We have modified accordingly to address these comments:

“Following this assessment, participants underwent a 30-minute rest period in a passive standing posture on both feet, during which all parameters potentially influencing autonomic function (hydration, restroom use, posture) were controlled and standardized. During this interval, 28 out of the 34 firefighters were randomly selected to be fitted with a heart rate monitor.”

L147. Why only 28 with HRV measurements? What were not involved volunteers doing during these 15 minutes?

Unfortunately, only 28 out of 34 participants could undergo HRV measurements because our equipment allowed a maximum of 28 sensors, and for operational reasons related to firefighters’ duty schedules, it was not possible to organize an additional complete experimental session. Nevertheless, since 34 volunteers had been successfully recruited, we decided to retain the six additional participants for the physical performance tests. During the HRV recording period, the non-equipped participants followed exactly the same procedures as the equipped ones.

L149. Figure 1 indicates 15 minutes for HRV test (also L154).

Yes, precisely. The standardized Schmitt Tilt Test (Schmitt et al., 2013) involves an 8-minute cardiac recording in the supine position followed by 7 minutes in the standing position, resulting in a total duration of 15 minutes.

L154-155. HRV was evaluated after a single warm-up while muscle force/endurance were evaluated after 2 warm-up protocols. Have authors tested the potential bias? Were muscle properties similarly affected after a single o

---

## [Decision Letter · Decision Letter 1]

9 Nov 2025

Mental imagery and breathing exercises integrated into a standardized warm-up routine enhance sympathetic activation and optimize muscular performance in firefighters

PONE-D-25-25807R1

Dear Dr. Amarantini,

We’re pleased to inform you that your manuscript has been judged scientifically suitable for publication and will be formally accepted for publication once it meets all outstanding technical requirements.

Kind regards,

Emiliano Cè, Ph.D.

Academic Editor

PLOS ONE

Additional Editor Comments (optional):

Reviewers' comments:

Reviewer's Responses to Questions

**Comments to the Author**

Reviewer #1: All comments have been addressed

2. Is the manuscript technically sound, and do the data support the conclusions?

Reviewer #1: Yes

3. Has the statistical analysis been performed appropriately and rigorously?

Reviewer #1: Yes

4. Have the authors made all data underlying the findings in their manuscript fully available?

Reviewer #1: Yes

5. Is the manuscript presented in an intelligible fashion and written in standard English?

Reviewer #1: Yes

Reviewer #1: The reviewer thanks the authors for the huge work achieved (answers and alterations). Authors have adequately modified the manuscript.

**Do you want your identity to be public for this peer review?** For information about this choice, including consent withdrawal, please see our Privacy Policy

Reviewer #1: No

---

## [Editor Report · Acceptance letter]

PONE-D-25-25807R1

PLOS ONE

Dear Dr. Amarantini,

I'm pleased to inform you that your manuscript has been deemed suitable for publication in PLOS ONE. Congratulations! Your manuscript is now being handed over to our production team.

Kind regards,

on behalf of

Prof. Emiliano Cè

Academic Editor

PLOS ONE